# Diversity for The Win: Towards Building Multi-Agent Systems with Heterogeneous LLMs

## Abstract

LLM-based multi-agent systems (MAS) extend the capabilities of single LLMs by enabling cooperation among multiple specialized agents. However, most existing MAS frameworks rely on a single LLM to drive all agents, constraining the system's intelligence to the limit of that model. This paper explores the paradigm of heterogeneous LLM-driven MAS, where agents are powered by diverse LLMs, elevating the system's potential to the collective intelligence of diverse LLMs. We introduce X-MAS-Bench, a comprehensive testbed designed to evaluate the performance of various LLMs across different domains and MAS-related functions. As an extensive empirical study, we assess 27 LLMs across 5 domains (encompassing 21 test sets) and 5 functions, conducting over 1.7 million evaluations to identify optimal model selections for each domain-function combination. Building on these findings, we demonstrate that transitioning from homogeneous to heterogeneous LLM-driven MAS can significantly enhance system performance without requiring structural redesign. Specifically, in a chatbot-only MAS scenario, the heterogeneous configuration yields up to 8.4% performance improvement on the MATH dataset. In a mixed chatbot-reasoner scenario, the heterogeneous MAS could achieve a remarkable 47% performance boost on the AIME dataset. Our results underscore the transformative potential of heterogeneous LLMs in MAS, highlighting a promising avenue for advancing scalable, collaborative AI systems.

## 1 Introduction

Large language models (LLMs) such as GPT OpenAI (2023), Gemini Team et al. (2024), Qwen Yang et al. (2024b), have been applied across various domains. However, despite their remarkable capabilities, LLMs often struggle with multifaceted, complex, and real-world problems due to inherent limitations such as hallucinations Zhang et al. (2023); Min et al. (2023).

In response to these limitations, LLM-based multi-agent systems (MAS) have emerged as a promising solution Ye et al. (2025); Qian et al. (2024); Gottweis et al. (2025). MAS involves the collaboration of multiple agents, each specialized in specific functions, to address problems more effectively than a single model could. his paradigm has been successfully applied across various scenarios, including software development Qian et al. (2024); Hong et al. (2024), mathematics Lei et al. (2024); Liu et al. (2024), and scientific discovery Boiko et al. (2023); Lu et al. (2024). For instance, ChatDev Qian et al. (2024), MetaGPT Hong et al. (2024), and EvoMAC Hu et al. (2025b) utilize multiple coding agents (e.g., coders and testers) to improve software programming, while AI co-scientist Gottweis et al. (2025) employs a MAS to enhance biomedical and scientific research.

Despite notable progress, most existing MAS frameworks rely on a single LLM to drive all agents Hong et al. (2024); Qian et al. (2024); Liu et al. (2024); Ye et al. (2025); Hu et al. (2025b); Du et al. (2024); Chen et al. (2024b). This manner inherently limits the systems intelligence to that of the underlying model. For example, if a single LLM produces fundamental errors in certain facts, these mistakes are unlikely to be corrected through the collaboration of agents powered by the same model. Inspired by the advantages of diversity in collective intelligence Hong & Page (2004); Kozhevnikov et al. (2014); Aggarwal et al. (2015), this paper explores MAS with heterogeneous LLMs (X-MAS), pushing the systems capabilities beyond its previous limit to harness the collective

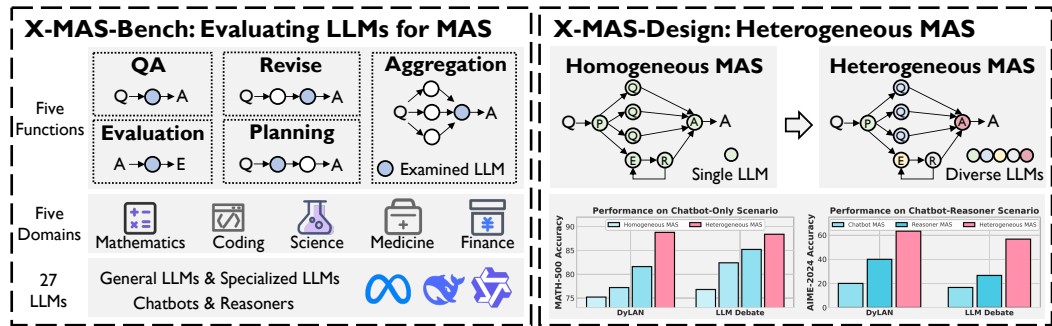

Figure 1: Overview of our X-MAS-Bench and X-MAS-Design. X-MAS-Bench assesses the capabilities of LLMs in MAS while X-MAS-Design focuses on transitioning a homogeneous MAS to a heterogeneous one, gaining from the observations in X-MAS-Bench. Experiments on chatbot-only and mixed chatbot-reasoner scenarios evidently show the benefits of heterogeneous MAS.

potential of LLMs trained on diverse corpora or by different teams Yang et al. (2024a); Dubey et al. (2024); Yang et al. (2024c).

To provide a comprehensive evaluation of LLMs in MAS, we introduce **X-MAS-Bench**, a testbed designed to assess the performance of various LLMs across different MAS-related functions and domains. Specifically, we consider 5 representative functions of agents in MAS, including question-answering Du et al. (2024); Hong et al. (2024), revise Hu et al. (2025b); Madaan et al. (2024), aggregation Qian et al. (2025); Liang et al. (2024), planning Islam et al. (2024); Lei et al. (2024), and evaluation Chen et al. (2024b); Qian et al. (2024); as well as 5 common domains, including mathematics, coding, science, medicine, and financespanning 21 test sets. Each function is assessed under controlled experimental conditions. For example, when assessing aggregation, each query is input into several pre-defined LLMs, whose outputs are concatenated to be aggregated by the examined LLM. The aggregated responses of various LLMs are then evaluated and compared. Finally, we assess 27 LLMs across these 5 functions and 5 domains, conducting over 1.7 million evaluations to identify the optimal model selections for each domain-function combination. Our findings include that (1) no single LLM excels across all scenarios, (2) a single LLM could have significant performance variation across functions and domains, (3) different LLMs may show large performance disparities within the same function and domain, (4) smaller LLMs can sometimes outperform larger ones, highlighting the potential advantages of employing heterogeneous LLMs in MAS. These results provide valuable insights for researchers and practitioners in selecting the most appropriate LLMs for their specific MAS applications.

Building on these observations, we explore the effects of transitioning from homogeneous to heterogeneous LLM-driven MAS (**X-MAS-Design**). As a proof of concept, given the implementation of a MAS method, we simply assign agents with appropriate LLMs (taking seconds) by referring observations in X-MAS-Bench. To validate our idea, we examine three existing MAS frameworksLLM-Debate Du et al. (2024), AgentVerse Chen et al. (2024b), and DyLAN Liu et al. (2024)as well as a prototype MAS designed by us, which incorporates all five functions in one system. Our analysis covers five domains with no sample overlap compared to X-MAS-Bench. In a chatbot-only scenario, we observe consistent improvements in performance for heterogeneous MAS over homogeneous configurations, achieving up to a 8.4% performance gain on the MATH Hendrycks et al. (2021b) benchmark. Interestingly, while reasoner-only MAS often underperforms relative to chatbot-only systems, combining chatbot and reasoner in a heterogeneous MAS leads to significant performance improvements. Specifically, in the competition-level AIME-2024 benchmark, AgentVerse Chen et al. (2024b) is improved from 20% to 50%, and DyLAN Liu et al. (2024) improved from 40% to 63%. Our further experiments reveal that increasing the number of candidate LLMs for heterogeneous MAS results in a monotonic performance improvement, reinforcing the value of LLM diversity in MAS. Based on our work, future research could explore more nuanced strategies for selecting and integrating LLMs in heterogeneous MAS; investigate the scalability and adaptability of heterogeneous MAS across different industries and other complex tasks.

Our contributions are as follows:

1. **X-MAS-Bench:** We assess 27 LLMs across 5 MAS-related functions and 5 domains, conducting over 1.7 million evaluations to identify diverse optimal model selections for each domain-function combination. These observations could benefit researchers and practitioners in building MAS.

2. **X-MAS-Design:** Based on findings in X-MAS-Bench, we propose to transition existing MAS methods from homogeneous to heterogeneous LLM-driven MAS. We conduct extensive experiments, showing that heterogeneous MAS consistently outperforms homogeneous MAS.

3. **Open Source:** We release all data, code, and evaluation results to facilitate future MAS research.

## 2 RELATED WORK

**LLM-based MAS.** LLM-based multi-agent systems (MAS) leverage multiple LLM-based agents to collaborate for better task solving than single LLM Chen et al. (2024b); Hong et al. (2024); Hu et al. (2025a); Ye et al. (2025). ChatDev Qian et al. (2024), MetaGPT Hong et al. (2024), and EvoMAC Hu et al. (2025b) use multiple coding agents (e.g., coders and testers) for software programming; while MACM Lei et al. (2024) applies math agents for mathematics. Focusing on general tasks, debate-based methods Du et al. (2024); Liang et al. (2024) enable multiple experts in debating for better solutions; AgentVerse Chen et al. (2024b) and DyLAN Liu et al. (2024) dynamically adjust the agent team for task solving; while MAS-GPT Ye et al. (2025) trains an LLM for generating MAS. However, all of these methods rely on a single LLM to drive all agents, which inherently limits the system's intelligence to that of the underlying LLM. This paper proposes to push the limit by harnessing the collective intelligence of heterogeneous LLMs from different sources.

**Heterogeneous LLMs.** In a general context of LLMs, there are several works related to using heterogeneous LLMs Chen et al. (2023); Venkatraman et al. (2024). LLM-Blender Jiang et al. (2023) trains a model for ensembling outputs from multiple LLMs. MoA Wang et al. (2025) and ReConcile Chen et al. (2024a) enable multiple LLMs for discussion, however, involving all candidate LLMs without considering their appropriateness. MASRouter Yue et al. (2025) manually selects several candidate LLMs for MAS and is optimized for their specific framework. In contrast, our paper systematically assess the capabilities of LLMs across several MAS-related functions and domains, aiming to universally benefit the design of heterogeneous MAS for various MAS methods.

**Benchmarking LLMs.** Many works benchmark the capabilities of LLMs in various domains (such as math Hendrycks et al. (2021b), coding Jimenez et al. (2024), science Rein et al. (2023), medicine OpenAI (2025), and finance Xie et al. (2023)) and functions (such as planning Valmeekam et al. (2023) and evaluation Tan et al. (2025)). However, our paper for the first time benchmarks LLMs for MAS, which assesses the capabilities of LLMs across 25 function-domain perspectives related to MAS.

## 3 X-MAS-BENCH: EVALUATING LLMS FOR MAS

X-MAS-Bench is a testbed designed to assess the performance of various LLMs across different MAS-related functions and domains. Specifically, we consider 5 representative functions of agents in MASquestion-answering Du et al. (2024); Hong et al. (2024), revise Hu et al. (2025b); Madaan et al. (2024), aggregation Qian et al. (2025); Liang et al. (2024), planning Islam et al. (2024); Lei et al. (2024), and evaluation Chen et al. (2024b); Qian et al. (2024). Orthogonally, we investigate behaviors in 5 domains, including mathematics, coding, science, medicine, and financespanning 21 test sets. Each function is assessed under controlled experimental conditions. In this section, we demonstrate the details of experimental conditions in Section 3.1 and experimental results in Section 3.2.

### 3.1 BENCHMARKING MAS-RELATED FUNCTIONS

To assess LLM capabilities in multi-agent systems (MAS), we decompose behaviors into five key functions: question-answering, revise, aggregation, planning, and evaluation. Each uses a standardized prompt protocol, with only the evaluated LLM varying. We detail each below.

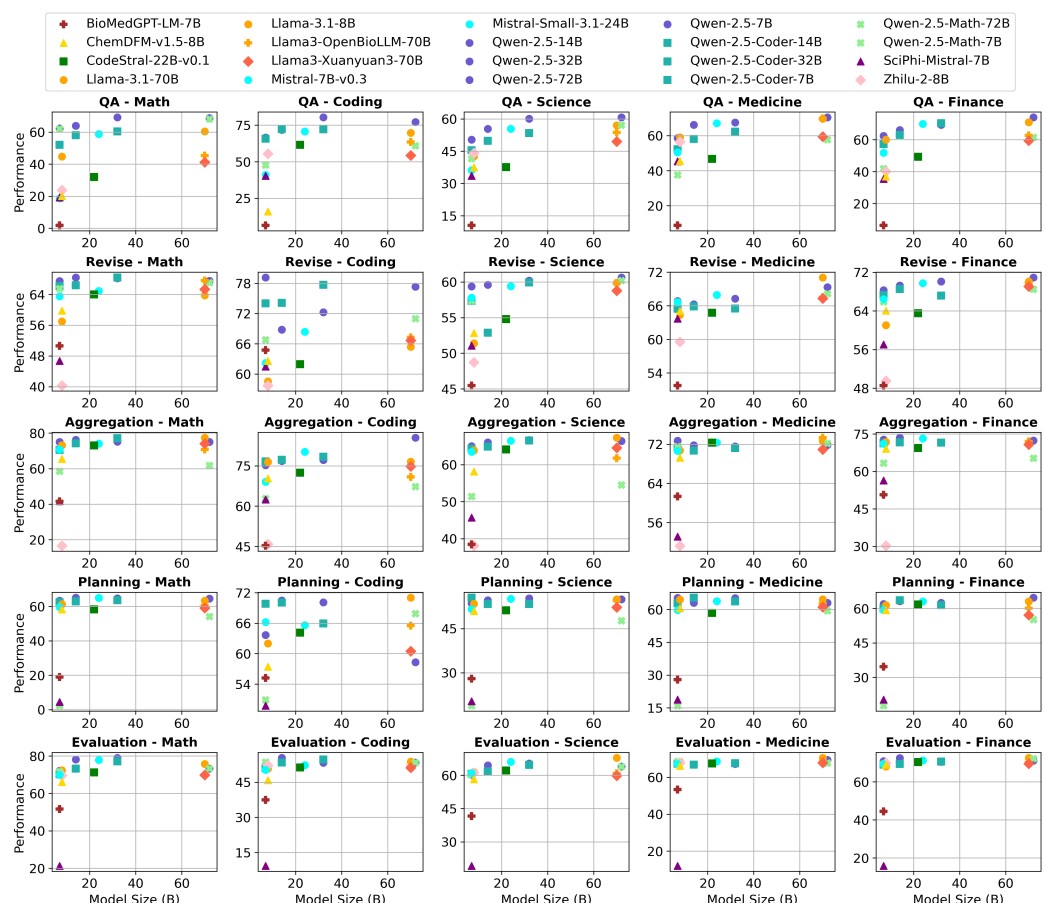

Figure 2: Benchmarking chatbot LLMs on 5 MAS-related functions and 5 domains. We see that no single LLM excels across all scenarios, indicating the potential advantages of employing heterogeneous LLMs in MAS. All evaluation results will be open-sourced for future research.

**Question-answering.** QA evaluates a LLMs ability to understand a query and generate a correct free-text answer. It underpins MAS like LLM-Debate Du et al. (2024) and MetaGPT Hong et al. (2024). Assessment involves inputting a sampled query (e.g., from MATH Hendrycks et al. (2021b)) and comparing outputs to ground-truth for accuracy.ă

**Revise.** Revise tests revising a potentially flawed initial answer into a corrected one, as in Evo-MAC Hu et al. (2025b) and Self-Refine Madaan et al. (2024). Given a query and pre-defined LLM-generated answer, the examined LLM reasons and revises to a final answer. Prompts are identical across LLMs; accuracy against ground-truth measures capability.

**Aggregation.** Aggregation combines multiple candidate answers into a coherent, improved final one, key in MacNet Qian et al. (2025) and MAD Liang et al. (2024). For each query, fixed responses from 3 pre-defined LLMs are provided in consistent format; the LLM synthesizes the final answer. Accuracy against ground-truth evaluates performance.

**Planning.** Planning decomposes tasks into sub-tasks and assigns agent roles for collaborative solving, as in MACM Lei et al. (2024) and MapCoder Islam et al. (2024). The LLM generates a plan with role descriptions and workflow in a predefined format. Extracted roles activate corresponding candidate LLMs; fixed inputs ensure fairness. Final task accuracy proxies planning ability.

**Evaluation.** Evaluation assesses critiquing other agents outputs for quality or correctness, used in AgentVerse Chen et al. (2024b) and ChatDev Qian et al. (2024). Presented with a query and pre-defined LLM answer, the LLM judges correctness. Consistent inputs enable fair comparison; judgments are scored against ground-truth.

Table 1: Summary of top-3 LLMs for each function-domain combination (chatbot-only scenario). All the assessed LLMs are instructed models (e.g., Qwen2.5-32B denotes Qwen2.5-32B-Instruct.). We see that no single LLM excels across all scenarios. Meanwhile, the top models are not always those with the largest sizes, indicating the potential of improving both performance and cost.

| Function | Rank | Mathematics | Coding | Science | Medicine | Finance |
|---|---|---|---|---|---|---|
| QA | 1 | Qwen2.5-32B (69.2) | Qwen2.5-32B (80.3) | Qwen2.5-72B (60.7) | Qwen2.5-72B (70.4) | Qwen2.5-72B (74.0) |
| | 2 | Qwen2.5-72B (68.8) | Qwen2.5-72B (77.1) | Qwen2.5-32B (60.0) | Llama3-OpenBioLLM-70B (69.7) | Qwen2.5-32B (71.0) |
| | 3 | Qwen2.5-Math-72B (68.2) | Qwen2.5-Coder-14B (72.3) | Qwen2.5-Math-72B (57.1) | Llama-3.1-70B (69.6) | Qwen2.5-Coder-32B (70.3) |
| Revise | 1 | Qwen2.5-Coder-32B (68.4) | Qwen2.5-7B (79.2) | Qwen2.5-72B (60.6) | Llama-3.1-70B (71.0) | Qwen2.5-72B (70.9) |
| | 2 | Qwen2.5-14B (68.4) | Qwen2.5-Coder-32B (77.7) | Qwen2.5-32B (60.2) | Qwen2.5-72B (69.3) | Llama-3.1-70B (70.1) |
| | 3 | Qwen2.5-32B (68.2) | Qwen2.5-7B (77.3) | Qwen2.5-Math-72B (60.2) | Qwen2.5-Math-72B (68.1) | Qwen2.5-32B (70.1) |
| Aggregation | 1 | Llama-3.1-70B (77.4) | Qwen2.5-72B (85.5) | Llama-3.1-70B (67.3) | Llama3-OpenBioLLM-70B (73.4) | Qwen2.5-14B (73.6) |
| | 2 | Qwen2.5-Coder-32B (77.1) | Mistral-Small-3.1-24B (80.2) | Qwen2.5-32B (66.7) | Qwen2.5-7B (72.7) | Mistral-Small-3.1-24B (73.2) |
| | 3 | Qwen2.5-14B (76.2) | Qwen2.5-Coder-32B (78.4) | Qwen2.5-Coder-32B (66.5) | Llama-3.1-70B (72.7) | Qwen2.5-7B (72.8) |
| Planning | 1 | Qwen2.5-14B (65.0) | Llama-3.1-70B (71.0) | Qwen2.5-Coder-7B (55.5) | Qwen2.5-Coder-14B (65.4) | Qwen2.5-72B (64.7) |
| | 2 | Mistral-Small-3.1-24B (65.0) | Qwen2.5-14B (70.5) | Qwen2.5-32B (55.3) | Qwen2.5-7B (65.3) | Qwen2.5-Coder-14B (63.6) |
| | 3 | Qwen2.5-32B (64.7) | Qwen2.5-32B (70.1) | Mistral-Small-3.1-24B (55.1) | Qwen2.5-32B (65.2) | Qwen2.5-14B (63.2) |
| Evaluation | 1 | Qwen2.5-32B (79.0) | Qwen2.5-14B (55.4) | Llama-3.1-70B (67.9) | Llama-3.1-70B (70.5) | Llama-3.1-70B (72.6) |
| | 2 | Qwen2.5-14B (78.1) | Qwen2.5-Coder-32B (54.7) | Mistral-Small-3.1-24B (66.1) | Qwen2.5-72B (69.4) | Qwen2.5-14B (72.6) |
| | 3 | Mistral-Small-3.1-24B (77.9) | Llama-3.1-70B (53.8) | Qwen2.5-32B (65.3) | Mistral-Small-3.1-24B (68.7) | Qwen2.5-Math-72B (72.3) |

## 3.2 Experiments in Evaluating LLMs across Functions and Domains

Following the above definitions of functions, this section assesses the capabilities of various LLMs in different functions and domains, aiming at demonstrating the landscape of LLMs for MAS. The reported results are expected to demonstrate the potential of leveraging heterogeneous LLMs for MAS and facilitate future researchers in choosing appropriate LLMs for their MAS.

**Experimental setups.** We examine 27 LLMs, covering 20 chatbots (i.e., instructed LLMs) and 7 reasoners (i.e., reasoning LLMs). Among the 20 chatbots, we consider general chatbots trained by different companies such as Llama Dubey et al. (2024), Qwen Yang et al. (2024a), Mistral Mistral (2024b; 2025), and domain-specific chatbots including mathematics Yang et al. (2024c), coding Hui et al. (2024), science Zhao et al. (2025); SciPhi (2023), medicine Duxiaoman-DI (2024); SYSU-MUCFC-FinTech-Research-Center (2024), and finance Duxiaoman-DI (2024); SYSU-MUCFC-FinTech-Research-Center (2024). The reasoners include LLMs from DeepSeek Guo et al. (2025), Qwen Team (2025b), and others Team (2025a). We set each models maximum token limit to its own capacity, 8192 tokens maxed, with a temperature of 0.5 by default. Specially, all LLMs instantiated within the planning workflow are executed with their temperature fixed at 0 to guarantee as the planning involves format-following. Our datasets cover domains including mathematics Hendrycks et al. (2021b); Ling et al. (2017); Gao et al. (2023); Maxwell-Jia (2024); Hendrycks et al. (2021a); Wang et al. (2024b), coding Chen et al. (2021a); Austin et al. (2021); Liu et al. (2023), science Rein et al. (2023); Wang et al. (2024a); Sun et al. (2024); Feng et al. (2024), medicine Pal et al. (2022); Jin et al. (2021; 2019), and finance Islam et al. (2023); Chen et al. (2021b); Malo et al. (2014), where each dataset is randomly sampled up to 500 examples without replacement; see more details in Section D.

**No single LLM excels across all scenarios.** We plot the size-performance values of each evaluated chatbot LLM across 25 function-domain combinations in Figure 2 and report the summary of top-3 LLMs for each combination in Table 1; see results of all LLMs in Figure 5 and Table 5. From these results, we see that (1) No single LLM excels universally across all scenarios. A heterogeneous MAS can capitalize on these differences by assigning scenario-specialized models (e.g., Llama3-OpenBioLLM for medicine) to specific agents, maximizing collective intelligence. (2) LLMs exhibit varied performance across MAS-related functions, reinforcing the value of heterogeneity.

**A single LLM could have significant performance variation across domains and functions.** Individual LLMs exhibit substantial performance disparities when evaluated across different domains and functions, underscoring the limitations of relying on a single model in a homogeneous MAS. For instance, in Figure 2, Qwen2.5-7B performs exceptionally well for revising in coding domain; while dropping to a mid-tier level for revising in medicine domain and planning in coding domain.

**There are large performance disparities across LLMs within the same domain and function.** For the function of revise or the domain of coding, we observe diverse behaviors on the examined LLMs, as shown by disperse scatters in Figure 2 (second row and second column).

**Smaller LLMs can outperform larger ones in niche scenarios.** While larger models like Qwen2.5-72B-Instruct and Llama-3.1-70B-Instruct often lead, smaller models occasionally excel in specific function-domain pairs. For example, in revise-coding pair, Qwen2.5-7B-Instruct (79.2) outperforms Qwen2.5-72B-Instruct (77.3); while in aggregation-finance and evaluation-finance pairs, Qwen2.5-14B achieves the best performance among all models. This indicates that heterogeneous MAS can incorporate smaller, specialized models to optimize performance and computational efficiency, reducing reliance on resource-intensive large models while maintaining or improving outcomes.

**Low-performing models highlight the risk of homogeneous MAS.** Some models consistently underperform across domains and functions (e.g., BioMedGPT-LM-7B and SciPhi-Mistral-7B-32k). A homogeneous MAS relying on such models would be severely limited, whereas a heterogeneous setup can mitigate this by integrating appropriate and high-performing LLMs.

**Consistent high performers enable robust heterogeneous configurations.** Models like Qwen-2.5-32B-Instruct, Qwen-2.5-72B-Instruct, and Llama-3.1-70B-Instruct frequently rank among the top across domains and functions (e.g., 80.3 in QA-coding, 79.0 in evaluation-math for Qwen-2.5-32B-Instruct). These models can serve as reliable anchors in a heterogeneous MAS, complemented by specialized models for niche tasks (e.g., Llama3-OpenBioLLM-70B in medicine), ensuring robust and scalable performance improvements.

## 4 X-MAS-DESIGN: LEVERAGING DIVERSITY FOR MAS

Based on the findings in X-MAS-Bench (Section 3.2), we explore the effects of transitioning from homogeneous to heterogeneous LLM-driven MAS (X-MAS-Design). We show how a homogeneous MAS is transformed into a heterogeneous MAS in Section 4.1. We provide experimental results in a chatbot-only scenario (Section 4.2) and a mixed chatbot-reasoner scenario (Section 4.3).

### 4.1 TRANSITIONING FROM HOMOGENEOUS TO HETEROGENEOUS LLM-DRIVEN MAS

**Transitioning existing MAS methods.** As a proof of concept, we aim to show that a simple manual modification of the LLM configurations can enhance the performance of MAS without any structural improvement. For each target MAS method (e.g., AgentVerse Chen et al. (2024b), LLM-Debate Du et al. (2024)), we retain the original agent roles and interaction topology but substitute the single homogeneous LLM with several appropriate LLMs for the agents. Concretely, for each domain-function pair in the original design (e.g., the evaluator for coding in AgentVerse), we replace the uniform LLM driver with the top performer in the pool of available models based on observations from X-MAS-Bench (Section 3.2). By preserving the method's interaction logic and prompt templates, we ensure that any performance gains stem solely from LLM heterogeneity rather than modifications of workflow. Please note that this modification is efficient as it only takes human researchers less than one minute to accomplish and could be automated even if we replace humans with LLMs with limited sizes (e.g., 7B Yang et al. (2024a)).

**X-MAS-Proto.** In addition to adapting existing MAS methods to heterogeneous ones, we implement XMASProto, a prototype MAS that explicitly implements all five functions (QA, revise, aggregation, planning, evaluation) in a single pipeline, serving as a proper object for investigation. The system (see the MAS in Figure 1) first invokes a planning agent to generate several different high-level ideas to the question; next, multiple QA agents concurrently answer the query based on its corresponding ideas while one of the answers will be evaluated and revised to obtain a potentially better answer; finally, an aggregation agent synthesizes across answers to get the final solution. With X-MAS-Proto, we could straightforwardly assign appropriate LLMs for different functional agents, aiming to clearly demonstrate the benefits of LLM heterogeneity in MAS.

### 4.2 EXPERIMENTS IN CHATBOT-ONLY SCENARIOS

**Experimental setups.** We experiment on X-MAS-Proto and three existing MAS methods including AgentVerse Chen et al. (2024b), LLM-Debate Du et al. (2024), and DyLAN Liu et al. (2024). Considering performances and efficiencies, we select four candidate chatbot LLMs: Qwen2.532B, MistralSmall3.124B, Qwen2.5Coder32B, and Qwen2.5Math7B. We test MAS on a *held-out* test

Table 2: Transitioning from homogeneous to heterogeneous LLM-driven MAS (X-MAS-Design). There are four considered MAS methods and four candidate models. X-MAS-Design consistently achieves top performances across 5 domains (3 are relatively out-of-domain for candidate LLMs).

| MAS Method | LLM | Math | Coding | Science | Medicine | Finance | Average |
|---|---|---|---|---|---|---|---|
| AgentVerse Chen et al. (2024b) | Qwen2.5-Math-7B | 2.40 | 3.21 | 0.40 | 6.00 | 5.33 | 3.47 |
| | Qwen2.5-Coder-32B | 75.20 | 72.69 | 32.00 | 47.60 | 64.00 | 58.30 |
| | Qwen2.5-32B | 83.20 | 76.31 | 34.00 | 50.40 | **74.67** | 63.72 |
| | Mistral-3.1-24B | 66.80 | 62.25 | 31.20 | 40.00 | 65.33 | 55.12 |
| | **X-MAS-Design** | **88.40** | **77.51** | **41.20** | **51.20** | 72.00 | **66.06** |
| LLM-Debate Du et al. (2024) | Qwen2.5-Math-7B | 79.20 | 40.96 | 29.60 | 35.20 | 30.67 | 43.13 |
| | Qwen2.5-Coder-32B | 82.40 | 78.71 | 34.40 | 46.80 | 68.00 | 62.06 |
| | Qwen2.5-32B | 85.20 | 75.50 | 32.80 | 50.80 | 77.33 | 64.33 |
| | Mistral-3.1-24B | 76.80 | 66.67 | 33.60 | **52.00** | 66.67 | 59.15 |
| | **X-MAS-Design** | **88.40** | **79.92** | **39.20** | 51.60 | **77.33** | **67.29** |
| DyLAN Liu et al. (2024) | Qwen2.5-Math-7B | 0.00 | 13.25 | 15.20 | 13.20 | 5.33 | 9.40 |
| | Qwen2.5-Coder-32B | 77.20 | 78.31 | 34.80 | 41.60 | 61.33 | 58.65 |
| | Qwen2.5-32B | 81.60 | 74.70 | 38.00 | 46.00 | 73.33 | 62.73 |
| | Mistral-3.1-24B | 75.20 | 61.85 | 32.80 | 41.60 | 72.00 | 56.69 |
| | **X-MAS-Design** | **88.80** | **78.71** | **38.80** | **47.20** | **76.00** | **65.90** |
| X-MAS-Proto | Qwen2.5-Math-7B | 10.40 | 12.85 | 2.00 | 10.80 | 5.33 | 8.28 |
| | Qwen2.5-Coder-32B | 82.00 | 76.71 | 33.60 | 46.80 | 58.67 | 59.56 |
| | Qwen2.5-32B | 82.00 | 69.88 | 31.20 | 45.60 | 72.00 | 60.14 |
| | Mistral-3.1-24B | 78.80 | 63.05 | 34.40 | 46.40 | 72.00 | 58.93 |
| | **X-MAS-Design** | **90.40** | **78.71** | **40.00** | 46.80 | 73.33 | **65.85** |

splits of MATH500, MBPP, SciBench, PubMedQA, and FinanceBench, covering the examined 5 domains. See model selection in Section E.

**Consistent performance gains of X-MAS-Design over homogeneous MAS.** Table 2 reports the performance comparisons of the homogeneous and heterogeneous versions of four MAS methods, where four LLMs are selected as candidates. The table demonstrates that X-MAS-Design, the heterogeneous MAS configuration, consistently outperforms all homogeneous configurations on average for four methods. In DyLAN, X-MAS-Design achieves an average performance of 65.90, surpassing the best homogeneous model (Qwen2.5-32B, 62.73) by 3 points. There are only two outlier casesLLM-Debate in medicine and Agentverse in financelikely due to the candidate LLMs not including specialized models for these particular domains. These results validate the X-MAS-Bench findings, which identified optimal model selections for domain-function combinations. By leveraging diverse and appropriate LLMs, X-MAS-Design harnesses collective intelligence, leading to superior performance without requiring structural changes to existing MAS methods.

**Method-agnostic benefits of heterogeneity.** The performance improvements of X-MAS-Design are consistent across all four MAS methods, despite their differing architectures and philosophies. This method-agnostic nature of X-MAS-Designs improvements highlights its versatility, providing strong evidence of our core idea in advocating X-MAS.

**X-MAS-Design could leverage the strengths of weak models to offset their weaknesses.** Homogeneous configurations show significant variability in performance across domains, with certain models underperforming in specific areas. For example, Qwen2.5-Math-7B performs poorly in most domains (e.g., 2.40 in Math, 0.40 in Science for AgentVerse), indicating its limited generalizability. Even stronger models like Qwen2.5-32B and Mistral-3.1-24B show weaknesses, such as Mistral-3.1-24Bs 31.2 in Science (AgentVerse) or Qwen2.5-32Bs 31.2 in Science (X-MAS-Proto). In contrast, X-MAS-Design consistently achieves balanced performance. That is, X-MAS-Design mitigates the limitations of individual LLMs by combining their strengths, indicating the benefits of collective intelligence and that our X-MAS-Bench provides helpful guidance for the design of X-MAS.

### 4.3 EXPERIMENTS IN MIXED CHATBOT-REASONER SCENARIOS

**Experimental setups.** The examined MAS methods follow that in Section 4.2. As chatbots and reasoners exhibit different behaviors, we consider two candidate LLMs: Qwen-2.5-72B-Instruct and DeepSeek-R1-Distill-Qwen-32B. These methods are tested on AIME-2024 and *held-out* splits

Table 3: Effectiveness of X-MAS-Design in mixing chatbots and reasoners. While reasoner-based homogeneous MAS performs worse than chatbot-based homogeneous MAS, incorporating chatbots and reasoners into heterogeneous MAS contributing to large performance improvement.

| MAS Method | LLM | Math | Coding | Science | Medicine | Finance | Average |
|---|---|---|---|---|---|---|---|
| AgentVerse Chen et al. (2024b) | Chatbot | 20.00 | 75.50 | 37.60 | 47.20 | 72.00 | 50.46 |
| | Reasoner | 0.00 | 11.65 | 5.60 | 44.40 | 21.33 | 16.60 |
| | **X-MAS-Design** | **50.00** | **77.91** | **40.00** | **52.40** | **78.67** | **59.80** |
| LLM-Debate Du et al. (2024) | Chatbot | 16.67 | 74.70 | 35.60 | 49.20 | 73.33 | 49.90 |
| | Reasoner | 26.67 | 79.12 | 41.60 | 50.00 | 72.00 | 53.88 |
| | **X-MAS-Design** | **56.67** | **81.12** | **44.40** | **54.40** | **80.00** | **63.32** |
| DyLAN Liu et al. (2024) | Chatbot | 20.00 | 74.70 | 34.00 | 44.00 | 70.76 | 48.67 |
| | Reasoner | 40.00 | 76.31 | 42.40 | 45.60 | 68.00 | 54.46 |
| | **X-MAS-Design** | **63.33** | **80.32** | **42.80** | **46.80** | **76.00** | **61.85** |
| X-MAS-Proto | Chatbot | 23.33 | 72.69 | 34.80 | 44.80 | 68.00 | 48.72 |
| | Reasoner | 0.00 | 71.49 | 23.20 | 49.20 | 56.00 | 39.98 |
| | **X-MAS-Design** | **70.00** | **79.12** | **47.20** | **52.80** | **76.00** | **65.02** |

of MBPP, SciBench, PubMedQA and FinanceBench, covering the five examined domains. We also test the methods on *entirely new* (compared to X-MAS-Bench) test sets: AIME-2025 OpenCompass (2024) (the latest AIME math competition) and MATH-MAS Zhou et al. (2025) (multi-step). See model selection in Section F.

**Mixing chatbots and reasoners in X-MAS-Design achieves superior performance across domains and MAS methods.** In Table 3, we explore the potential of mixing chatbot and reasoner LLMs in X-MAS-Design. From the table, we see that (1) X-MAS-Design, combining chatbot and reasoner agents powered by heterogeneous LLMs, consistently outperforms both standalone chatbot and reasoner configurations across all five domains. (2) Standalone chatbot and reasoner configurations show complementary strengths and weaknesses. The heterogeneous X-MAS-Design mitigates individual role limitations by combining chatbot and reasoner strengths, as guided by X-MAS-Benchs 1.7 million evaluations. This synergy enables robust performance across diverse domains.

**Mixing chatbots and reasoners leads to dramatic improvements in math domain (AIME).** We additionally evaluate homogeneous and heterogeneous MAS on two entirely new benchmarks: AIME-2025 and MATH-MAS in Table 4. From Table 3 and 4, we see that in math domain (i.e., AIME-2024, AIME-2025, MATH-MAS), X-MAS-Design contributes to substantial performance boosts. Notably, for X-MAS-Proto, X-MAS-Design scores 70% in AIME-2024, a 46.67%-point gain over the second-best homogeneous MAS, indicat-

Table 4: Examination on *entirely new* benchmarks. X-MAS-Design achieves significantly best performance.

| Benchmark | AIME-25 | MATH-M |
|---|---|---|
| Chatbot | 13.33 | 14.18 |
| Reasoner | 10.00 | 5.97 |
| **X-MAS-Design** | **46.67** | **48.13** |

ing the potential of X-MAS in reasoning-intensive tasks. Meanwhile, X-MAS-Design outperforms the second-best chatbot-based homogeneous MAS by 33% and 34% on the challenging AIME-2025 and MATH-MAS, respectively, indicating the generalization of our core idea. In the era where reasoning models prevail, our experiments point out a potential direction: further scaling compute with X-MAS that mixes chatbots and reasoners.

## 4.4 ABLATION STUDY

**Increasing the number of candidate models enhances the performance of X-MAS-Design.** Following the setup in Section 4.2, we conduct experiments with X-MAS-Proto on three domains (math, coding, and science) by tuning the number of candidate models. We use the full split for larger sample numbers. From Figure 3, we observe that (1) X-MAS-Design consistently outperforms homogeneous MAS (i.e., 1 candidate model), indicating the benefits of X-MAS. (2) With the number of candidate models increases, we can generally observe an increase of performance. One exception is in the science domain, which can be attributed that the added model from 2 to 3 is not closely related to science. This curve strongly indicates the benefits of including diverse LLMs in MAS.

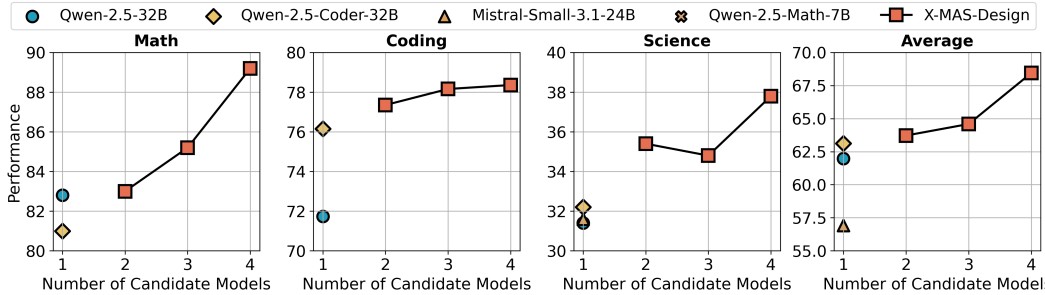

Figure 3: Diversity for the win. Experiments are conducted with X-MAS-Proto on three domains. Increasing the number of candidate models generally enhances the system performance, strongly indicating the benefits of LLM heterogeneity for MAS.

**Arbitrary model selection could lead to sub-optimal performance: X-MAS-Bench offers critical observations to guide the design of X-MAS.** To verify the effectiveness of the observations from X-MAS-Bench, we compare X-MAS with LLM selection guided by X-MAS-Bench to X-MAS with arbitrary selection. We follow the setup in Section 2, where we experiment on X-MAS-Proto on MATH-500. We arbitrarily determine 5 reasonable sets of configurations for designing X-MAS (see details in Section E.2), denoted by blue bars in Figure 4. Homogeneous MAS driven by three different LLMs is denoted by red bars. From the figure, we see that (1) X-MAS-Design, which is designed based on observations from X-MAS-

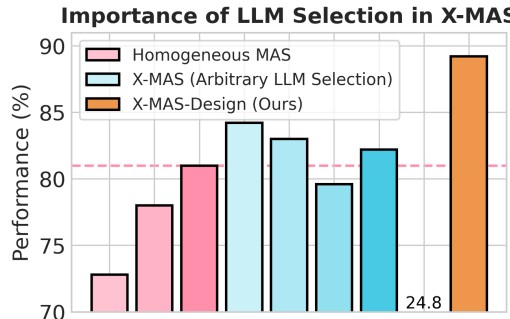

Figure 4: Comparing X-MAS with LLM selection guided by X-MAS-Bench and arbitrary selection. X-MAS-Design, which is guided by X-MAS-Bench, significantly performs the best.

Bench, significantly performs the best. (2) Among those 5 X-MAS without X-MAS-Bench's guidance, 3 of them achieve slightly better performance than homogeneous MAS, while 1 performs slightly worse than the best homogeneous MAS and 1 even performs significantly worst (only 24.8%). This indicates that appropriate LLM selection is critical for ensuring the performance of X-MAS and that results in X-MAS-Bench can provide valuable insights.

## 5 CONCLUSIONS

This paper advocates building LLM-based MAS with heterogeneous LLMs. We introduce X-MAS-Bench, a comprehensive testbed designed to assess the capabilities of various LLMs in supporting for MAS. We provide a systematic empirical study, which assesses 27 LLMs (both chatbots and reasoners, both genralists and specialists) across 5 representative MAS-related functions and 5 common domains, highlighting the potential of employing heterogeneous LLMs in MAS. Based on the insights from X-MAS-Bench, we examine the effects of transitioning from homogeneous to heterogeneous LLM-driven MAS (X-MAS-Design). Our experiments operating on 4 MAS methods demonstrate that the performance of MAS can be significantly and consistently improved by leveraging heterogeneous MAS without any structural re-design, strongly supporting our advocacy. See limitations in Section B.

Our work highlights an intriguing direction that leverages the collective intelligence of heterogeneous LLMs to achieve higher-level intelligence without additional training. Looking ahead, future research could explore areas such as automated or dynamic model selection, the impact of further scaling model candidates, optimizing the synergy between LLM selection and MAS, achieving strong performance with weaker agents, and training agents specifically suited for MAS.

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

## A    LLMs Usage

In this work, we employed GPT-4 exclusively during the writing phase to enhance readability and linguistic fluency through targeted polishing of the manuscript. Importantly, the LLM was not utilized for generating core scientific content, such as hypotheses, methodologies, results, or analyses, which were entirely developed by the authors. All outputs from the LLM underwent thorough human review, revision, and validation to ensure alignment with our original intent and academic integrity. We assume full responsibility for every aspect of the contributions, including those refined with LLM assistance.

## B    Limitations

Despite being the most comprehensive evaluation of LLMs for MAS, there are still LLMs that have not been included yet. When transitioning from homogeneous to heterogeneous MAS, we currently rely on manual modification as a proof of concept. Despite that the modification is quite simple, it is worthwhile to explore automated solutions.

## C    Broader Impacts

This paper introduces X-MAS-Bench, aiming at assessing the capabilities of LLMs when being incorporated in MAS. The assessed results and the corresponding findings could serve the community, facilitating appropriate model selections during the design of MAS. Our X-MAS-Design aims to transition an existing homogeneous MAS to a heterogeneous one. Similar concept could be extended to many existing MAS, making the overall system perform better.

The potential negative impacts of our approach mirror those associated with LLMs including risks of misuse. However, these issues are intrinsic to LLM usage in general.

## D    Experimental Setups

We examine 27 LLMs, covering 20 chatbots (i.e., instructed LLMs) and 7 reasoners (i.e., reasoning LLMs). Among the 20 chatbots, we consider general chatbots trained by different companies: Llama (Llama-3.1-8/70B-Instruct Dubey et al. (2024)), Qwen (Qwen2.5-7/14/32/72B-Instruct Yang et al. (2024a), Mistral (Mistral-7B-Instruct-v0.3 Mistral (2024b), Mistral-Small-3.1-24B-Instruct-2503 Mistral (2025)); we also include domain-specific chatbots including mathematics (Qwen2.5-Math-7/72B-Instruct Yang et al. (2024c)), coding (Qwen2.5-Coder-7/14/32B-Instruct Hui et al. (2024), Codestral-22B-v0.1 Mistral (2024a)), science (ChemDFM-v1.5-8B Zhao et al. (2025), SciPhi-Mistral-7B-32k SciPhi (2023)), medicine (Llama3-OpenBioLLM-70B Ankit Pal (2024) and BioMedGPT-LM-7B Luo et al. (2023)), and finance (Llama3-XuanYuan3-70B-Chat Duxiaoman-DI (2024) and ZhiLu-2-8B-Instruct SYSU-MUCFC-FinTech-Research-Center (2024)) The reasoners include LLMs from DeepSeek (DeepSeek-R1-Distill-Llama-8/70B and DeepSeek-R1-Distill-Qwen-7/14/32B Guo et al. (2025)), Qwen (QwQ-32B Team (2025b)) , other (OpenThinker-32B Team (2025a)) LLMs. We set each models maximum token limit to its own capacity, 8192 tokens maxed, with a temperature of 0.5 by default. Specially, all LLMs instantiated within the planning workflow are executed with their temperature fixed at 0 to guarantee as the planning involves format-following. Our datasets cover domains including mathematics (AIME-2024 Maxwell-Jia (2024), AQUA-RAT Ling et al. (2017), GSM-Hard Gao et al. (2023), MATH Hendrycks et al. (2021b), MMLU-Math Hendrycks et al. (2021a), MMLU-Pro-Math Wang et al. (2024b)), coding (HumanEval Chen et al. (2021a), HumanEval-Plus Liu et al. (2023), MBPP Austin et al. (2021), MBPP-Plus, MMLU-Coding, MMLU-Pro-coding), science (GPQA-Main Rein et al. (2023), GPQA-Diamond, SciBench Wang et al. (2024a), SciEval Sun et al. (2024), SciKnowEval Feng et al. (2024), MMLU-Sci, MMLU-Pro-Sci), medicine (MedMCQA Pal et al. (2022), MedQA Jin et al. (2021), PubMedQA Jin et al. (2019), MMLU-Med, MMLU-Pro-Med), and finance (FinanceBench Islam et al. (2023), FinQA Chen et al. (2021b), FPB Malo et al. (2014), MMLU-Finan, MMLU-Pro-Finan), where each dataset is randomly sampled up to 500 examples without replacement (except for SciKnowEval, from which we draw 800 instances to ensure sufficient coverage of its specialized tasks).

# E    EXPERIMENTS ON X-MAS IN CHATBOT-ONLY SCENARIOS

## E.1    EXPERIMENTS SETUPS OF X-MAS-DESIGN IN CHATBOT-ONLY SCENARIOS

The available LLMs are Qwen-2.5-32B-Instruct, Qwen-2.5-Coder-32B-Instruct, Qwen-2.5-Math-7B-Instruct and Mistral-Small-3.1-24B-Instruct-2503.

### E.1.1    AGENTVERSE

**Mathematics.**  The role assigner is Qwen-2.5-32B-Instruct, the solver is Qwen-2.5-Coder-32B-Instruct and Qwen-2.5-Math-7B-Instruct, the critic is Qwen-2.5-Coder-32B-Instruct, the evaluator is Qwen-2.5-32B-Instruct.

**Coding.**  The role assigner is Qwen-2.5-32B-Instruct, the solver is Qwen-2.5-32B-Instruct, the critic is Qwen-2.5-Coder-32B-Instruct, the evaluator is Qwen-2.5-Coder-32B-Instruct.

**Science.**  The role assigner is Qwen-2.5-32B-Instruct, the solver is Qwen-2.5-32B-Instruct, the critic is Qwen-2.5-32B-Instruct, the evaluator is Mistral-Small-3.1-24B-Instruct-2503.

**Medicine.**  The role assigner is Qwen-2.5-32B-Instruct, the solver is Qwen-2.5-32B-Instruct, the critic is Mistral-Small-3.1-24B-Instruct-2503, the evaluator is Mistral-Small-3.1-24B-Instruct-2503.

**Finance.**  The role assigner is Mistral-Small-3.1-24B-Instruct-2503, the solver is Qwen-2.5-Coder-32B-Instruct, the critic is Qwen-2.5-32B-Instruct, the evaluator is Mistral-Small-3.1-24B-Instruct-2503.

### E.1.2    LLM-DEBATE

**Mathematics.** The debate agent is Qwen-2.5-Coder-32B-Instruct and Qwen-2.5-Math-7B-Instruct, the aggregator is Mistral-Small-3.1-24B-Instruct-2503.

**Coding.**  The debate agent is Qwen-2.5-Coder-32B-Instruct, the aggregator is Mistral-Small-3.1-24B-Instruct-2503.

**Science.** The debate agent is Qwen-2.5-32B-Instruct, the aggregator is Qwen-2.5-32B-Instruct.

**Medicine.**  The debate agent is Qwen-2.5-32B-Instruct, the aggregator is Mistral-Small-3.1-24B-Instruct-2503.

**Finance.**  The debate agent is Qwen-2.5-Coder-32B-Instruct, the aggregator is Mistral-Small-3.1-24B-Instruct-2503.

### E.1.3    DYLAN

**Mathematics.**  The node agent is Qwen-2.5-Coder-32B-Instruct and Qwen-2.5-Math-7B-Instruct, the ranker is Mistral-Small-3.1-24B-Instruct-2503.

**Coding.**  The node agent is Qwen-2.5-Coder-32B-Instruct, the ranker is Mistral-Small-3.1-24B-Instruct-2503.

**Science.** The node agent is Qwen-2.5-32B-Instruct, the ranker is Qwen-2.5-32B-Instruct.

**Medicine.** The node agent is Mistral-Small-3.1-24B-Instruct-2503, the ranker is Mistral-Small-3.1-24B-Instruct-2503.

**Finance.**  The node agent is Qwen-2.5-Coder-32B-Instruct, the ranker is Mistral-Small-3.1-24B-Instruct-2503.

### E.1.4    X-MAS-PROTO

**Mathematics.** The planner is Qwen-2.5-32B-Instruct, the solver is Qwen-2.5-Coder-32B-Instruct and Qwen-2.5-Math-7B-Instruct, the reviser is Qwen-2.5-Coder-32B-Instruct, the evaluator is Qwen-2.5-32B-Instruct, the aggregator is Mistral-Small-3.1-24B-Instruct-2503.

**Coding.** The planner is Qwen-2.5-32B-Instruct, the solver is Qwen-2.5-Coder-32B-Instruct, the reviser is Qwen-2.5-Coder-32B-Instruct, the evaluator is Qwen-2.5-32B-Instruct, the aggregator is Mistral-Small-3.1-24B-Instruct-2503.

**Science.** The planner is Qwen-2.5-32B-Instruct, the solver is Qwen-2.5-32B-Instruct, the reviser is Qwen-2.5-32B-Instruct, the evaluator is Mistral-Small-3.1-24B-Instruct-2503, the aggregator is Qwen-2.5-32B-Instruct.

**Medicine.** The planner is Qwen-2.5-32B-Instruct, the solver is Qwen-2.5-32B-Instruct, the reviser is Mistral-Small-3.1-24B-Instruct-2503, the evaluator is Mistral-Small-3.1-24B-Instruct-2503, the aggregator is Mistral-Small-3.1-24B-Instruct-2503.

**Finance.** The planner is Mistral-Small-3.1-24B-Instruct-2503, the solver is Qwen-2.5-Coder-32B-Instruct, the reviser is Qwen-2.5-32B-Instruct, the evaluator is Mistral-Small-3.1-24B-Instruct-2503, the aggregator is Mistral-Small-3.1-24B-Instruct-2503.

### E.2 EXPERIMENTAL SETUPS OF X-MAS WITH NON-X-MAS-BENCH-GUIDED MODEL SELECTIONS

We arbitrarily determine five reasonable manually designed model configurations to examine the robustness and performance sensitivity of the X-MAS-Design under diverse agent choices. These configurations are constructed without referring to the X-MAS-Bench, and are denoted as X-MAS1 through X-MAS5. Each configuration includes distinct combinations of planner, solver, evaluator, reviser, and aggregator roles. For comparison, we also include the original X-MAS-Design configuration guided by X-MAS-Bench selection.

The X-MAS-Bench-guided configuration, referred to as **X-MAS-Design** in chatbot-only scenarios, adopts the following models for each agent role:

- **Planner**: Qwen-2.5-32B-Instruct
- **Solver**: Qwen-2.5-Coder-32B-Instruct
- **Evaluator**: Qwen-2.5-32B-Instruct
- **Reviser**: Qwen-2.5-Coder-32B-Instruct
- **Aggregator**: Mistral-Small-3.1-24B-Instruct-2503

This configuration reflects a well-balanced assignment with domain-specialized solvers (e.g., math) and stronger general-purpose planning and evaluation agents.

In contrast, the five alternative configurations (X-MAS1 to X-MAS5) are constructed based on general instruction-tuned LLMs without prior empirical optimization. These setups are:

**X-MAS1**

- **Planner**: Mistral-Small-3.1-24B-Instruct-2503
- **Solver**: Qwen-2.5-Math-7B-Instruct
- **Evaluator**: Qwen-2.5-Coder-32B-Instruct
- **Reviser**: Qwen-2.5-Math-7B-Instruct
- **Aggregator**: Qwen-2.5-32B-Instruct

**X-MAS2**

- **Planner**: Mistral-Small-3.1-24B-Instruct-2503
- **Solver**: Qwen-2.5-Coder-32B-Instruct
- **Evaluator**: Qwen-2.5-Math-7B-Instruct
- **Reviser**: Qwen-2.5-Coder-32B-Instruct
- **Aggregator**: Qwen-2.5-32B-Instruct

**X-MAS3**

- **Planner**: Qwen-2.5-Math-7B-Instruct
- **Solver**: Mistral-Small-3.1-24B-Instruct-2503
- **Evaluator**: Qwen-2.5-32B-Instruct
- **Reviser**: Mistral-Small-3.1-24B-Instruct-2503
- **Aggregator**: Qwen-2.5-Coder-32B-Instruct

**X-MAS4**

- **Planner**: Qwen-2.5-Coder-32B-Instruct
- **Solver**: Qwen-2.5-32B-Instruct
- **Evaluator**: Qwen-2.5-Math-7B-Instruct
- **Reviser**: Qwen-2.5-32B-Instruct
- **Aggregator**: Mistral-Small-3.1-24B-Instruct-2503

**X-MAS5**

- **Planner**: Qwen-2.5-32B-Instruct
- **Solver**: Qwen-2.5-Coder-32B-Instruct
- **Evaluator**: Mistral-Small-3.1-24B-Instruct-2503
- **Reviser**: Qwen-2.5-Coder-32B-Instruct
- **Aggregator**: Qwen-2.5-Math-7B-Instruct

All configurations are evaluated on the MATH-500 subset following the X-MAS-Proto scheme. The goal of this analysis is to understand the effect of heterogeneous agent assignments on final multi-agent performance, as well as to validate the necessity and advantages of X-MAS-Bench-guided agent selection. These baselines also serve to demonstrate the variance among manually configured pipelines in the absence of systematic design guidance.

# F EXPERIMENTS ON X-MAS IN MIXED CHATBOT-REASONER SCENARIOS

## F.1 MODEL SELECTIONS

The available LLMs are Qwen-2.5-72B-Instruct and DeepSeek-R1-Distill-Qwen-32B.

### F.1.1 AGENTVERSE

**Mathematics.** The role assigner is Qwen-2.5-72B-Instruct, the solver is DeepSeek-R1-Distill-Qwen-32B, the critic is DeepSeek-R1-Distill-Qwen-32B, the evaluator is DeepSeek-R1-Distill-Qwen-32B.

**Coding.** The role assigner is Qwen-2.5-72B-Instruct, the solver is DeepSeek-R1-Distill-Qwen-32B, the critic is DeepSeek-R1-Distill-Qwen-32B, the evaluator is DeepSeek-R1-Distill-Qwen-32B.

**Science.** The role assigner is Qwen-2.5-72B-Instruct, the solver is DeepSeek-R1-Distill-Qwen-32B, the critic is DeepSeek-R1-Distill-Qwen-32B, the evaluator is DeepSeek-R1-Distill-Qwen-32B.

**Medicine.** The role assigner is Qwen-2.5-72B-Instruct, the solver is Qwen-2.5-72B-Instruct, the critic is Qwen-2.5-72B-Instruct, the evaluator is DeepSeek-R1-Distill-Qwen-32B.

**Finance.** The role assigner is Qwen-2.5-72B-Instruct, the solver is DeepSeek-R1-Distill-Qwen-32B, the critic is DeepSeek-R1-Distill-Qwen-32B, the evaluator is DeepSeek-R1-Distill-Qwen-32B.

### F.1.2 LLM-DEBATE

**Mathematics.** The debate agent is DeepSeek-R1-Distill-Qwen-32B, the aggregator is DeepSeek-R1-Distill-Qwen-32B.

**Coding.** The debate agent is DeepSeek-R1-Distill-Qwen-32B, the aggregator is Qwen-2.5-72B-Instruct.

**Science.** The debate agent is DeepSeek-R1-Distill-Qwen-32B, the aggregator is DeepSeek-R1-Distill-Qwen-32B.

**Medicine.** The debate agent is Qwen-2.5-72B-Instruct, the aggregator is DeepSeek-R1-Distill-Qwen-32B.

**Finance.** The debate agent is DeepSeek-R1-Distill-Qwen-32B, the aggregator is DeepSeek-R1-Distill-Qwen-32B.

### F.1.3 DYLAN

**Mathematics.** The node agent is DeepSeek-R1-Distill-Qwen-32B, the ranker is Qwen-2.5-72B-Instruct.

**Coding.** The node agent is DeepSeek-R1-Distill-Qwen-32B, the ranker is Qwen-2.5-72B-Instruct.

**Science.** The node agent is DeepSeek-R1-Distill-Qwen-32B, the ranker is Qwen-2.5-72B-Instruct.

**Medicine.** The node agent is Qwen-2.5-72B-Instruct, the ranker is Qwen-2.5-72B-Instruct.

**Finance.** The node agent is DeepSeek-R1-Distill-Qwen-32B, the ranker is Qwen-2.5-72B-Instruct.

### F.1.4 X-MAS-PROTO

**Mathematics.** The planner is Qwen-2.5-72B-Instruct, the solver is DeepSeek-R1-Distill-Qwen-32B, the reviser is DeepSeek-R1-Distill-Qwen-32B, the evaluator is DeepSeek-R1-Distill-Qwen-32B, the aggregator is DeepSeek-R1-Distill-Qwen-32B.

**Coding.** The planner is Qwen-2.5-72B-Instruct, the solver is DeepSeek-R1-Distill-Qwen-32B, the reviser is DeepSeek-R1-Distill-Qwen-32B, the evaluator is DeepSeek-R1-Distill-Qwen-32B, the aggregator is Qwen-2.5-72B-Instruct.

**Science.** The planner is Qwen-2.5-72B-Instruct, the solver is DeepSeek-R1-Distill-Qwen-32B, the reviser is DeepSeek-R1-Distill-Qwen-32B, the evaluator is DeepSeek-R1-Distill-Qwen-32B, the aggregator is DeepSeek-R1-Distill-Qwen-32B.

**Medicine.** The planner is Qwen-2.5-72B-Instruct, the solver is Qwen-2.5-72B-Instruct, the reviser is Qwen-2.5-72B-Instruct, the evaluator is DeepSeek-R1-Distill-Qwen-32B, the aggregator is DeepSeek-R1-Distill-Qwen-32B.

**Finance.** The planner is Qwen-2.5-72B-Instruct, the solver is DeepSeek-R1-Distill-Qwen-32B, the reviser is DeepSeek-R1-Distill-Qwen-32B, the evaluator is DeepSeek-R1-Distill-Qwen-32B, the aggregator is DeepSeek-R1-Distill-Qwen-32B.

Table 5: Top-3 Models per Function and Domain (reasoner and chatbot)

| Function | Rank | Mathematics | Coding | Science | Medicine | Finance |
|---|---|---|---|---|---|---|
| QA | 🥇 | QwQ-32B (80.5) | DeepSeek-R1-Distill-Qwen-14B (82.0) | QwQ-32B (69.4) | DeepSeek-R1-Distill-Llama-70B (75.1) | DeepSeek-R1-Distill-Qwen-32B (74.8) |
| | 🥈 | DeepSeek-R1-Distill-Qwen-32B (79.0) | Qwen2.5-32B (80.3) | DeepSeek-R1-Distill-Llama-70B (69.4) | QwQ-32B (73.8) | QwQ-32B (74.6) |
| | 🥉 | DeepSeek-R1-Distill-Qwen-14B (78.8) | DeepSeek-R1-Distill-Qwen-32B (80.0) | DeepSeek-R1-Distill-Qwen-32B (68.3) | Qwen2.5-72B (70.4) | DeepSeek-R1-Distill-Llama-70B (74.3) |
| Revise | 🥇 | QwQ-32B (78.6) | DeepSeek-R1-Distill-Llama-70B (81.7) | QwQ-32B (67.0) | Llama-3.1-70B (71.0) | QwQ-32B (76.6) |
| | 🥈 | DeepSeek-R1-Distill-Llama-70B (78.2) | DeepSeek-R1-Distill-Qwen-32B (81.0) | DeepSeek-R1-Distill-Llama-70B (66.3) | DeepSeek-R1-Distill-Llama-70B (66.3) | DeepSeek-R1-Distill-Llama-70B (73.9) |
| | 🥉 | DeepSeek-R1-Distill-Qwen-32B (77.8) | Qwen2.5-7B (79.2) | DeepSeek-R1-Distill-Qwen-32B (65.9) | DeepSeek-R1-Distill-Llama-70B (70.7) | DeepSeek-R1-Distill-Qwen-32B (73.5) |
| Aggregation | 🥇 | QwQ-32B (83.2) | Qwen2.5-72B (85.5) | DeepSeek-R1-Distill-Llama-70B (71.7) | DeepSeek-R1-Distill-Llama-8B (74.1) | DeepSeek-R1-Distill-Qwen-32B (76.4) |
| | 🥈 | DeepSeek-R1-Distill-Qwen-32B (82.2) | QwQ-32B (84.2) | QwQ-32B (71.3) | DeepSeek-R1-Distill-Llama-70B (73.6) | DeepSeek-R1-Distill-Qwen-32B (76.4) |
| | 🥉 | DeepSeek-R1-Distill-Qwen-14B (81.2) | DeepSeek-R1-Distill-Llama-70B (83.1) | DeepSeek-R1-Distill-Qwen-32B (70.3) | DeepSeek-R1-Distill-Llama-70B (73.6) | QwQ-32B (74.6) |
| Planning | 🥇 | Qwen2.5-14B (65.0) | Llama-3.1-70B (71.0) | Qwen2.5-Coder-7B (56.1) | Qwen2.5-Coder-14B (65.4) | Qwen2.5-72B (64.7) |
| | 🥈 | Mistral-Small-3.1-24B (65.0) | Qwen2.5-14B (70.5) | Qwen2.5-32B (55.6) | Qwen2.5-7B (65.3) | Qwen2.5-Coder-14B (63.6) |
| | 🥉 | Qwen2.5-32B (64.7) | Qwen2.5-32B (70.1) | Qwen2.5-72B (55.6) | Qwen2.5-32B (65.2) | Qwen2.5-14B (63.2) |
| Evaluation | 🥇 | DeepSeek-R1-Distill-Llama-70B (85.9) | DeepSeek-R1-Distill-Qwen-32B (56.2) | DeepSeek-R1-Distill-Llama-70B (70.9) | Llama-3.1-70B (70.5) | OpenThinker-32B (76.6) |
| | 🥈 | QwQ-32B (84.2) | Qwen2.5-14B (55.4) | DeepSeek-R1-Distill-Qwen-32B (69.1) | DeepSeek-R1-Distill-Llama-70B (70.2) | QwQ-32B (73.8) |
| | 🥉 | OpenThinker-32B (83.3) | QwQ-32B (55.3) | OpenThinker-32B (69.0) | DeepSeek-R1-Distill-Qwen-14B (69.8) | DeepSeek-R1-Distill-Llama-70B (73.1) |

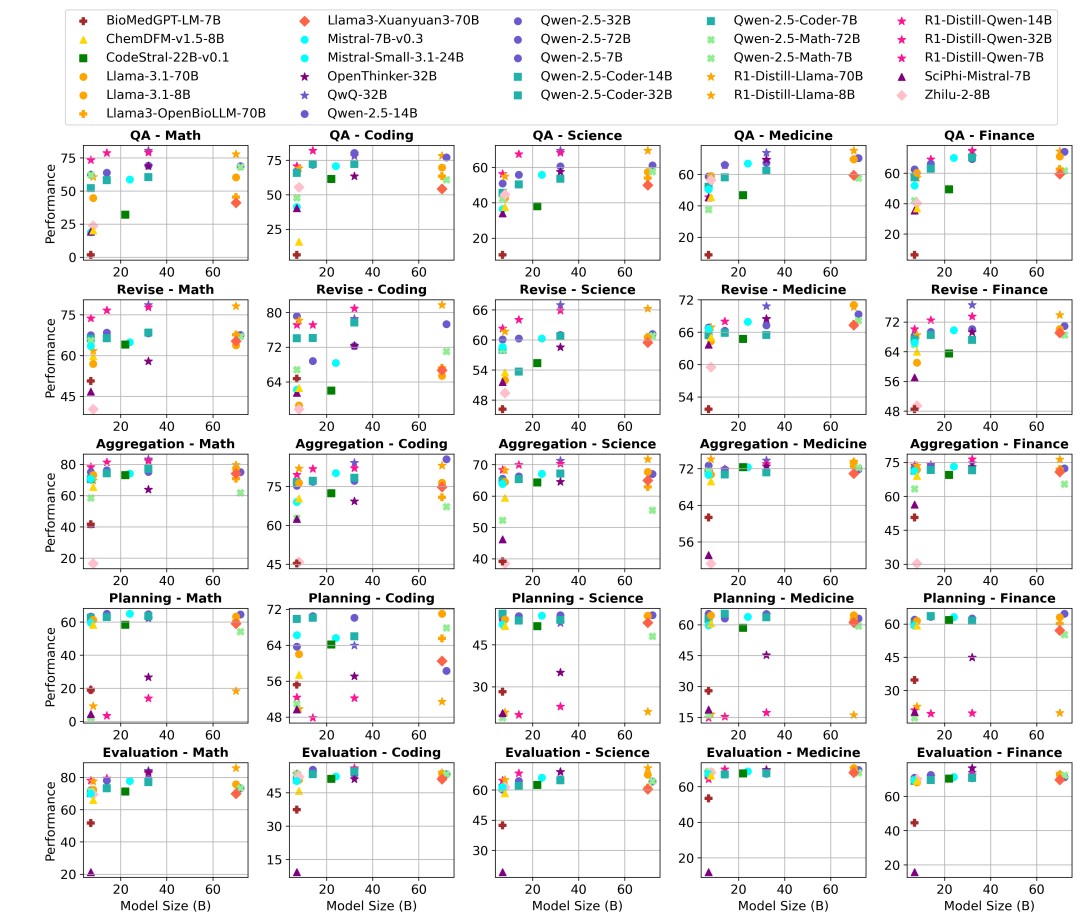

Figure 5: Benchmarking LLMs on 5 MAS-related functions and 5 domains.

Our datasets cover domains including mathematics (AIME-2024 Maxwell-Jia (2024), AQUA-RAT Ling et al. (2017), GSM-Hard Gao et al. (2023), MATH Hendrycks et al. (2021b), MMLU-Math Hendrycks et al. (2021a), MMLU-Pro-Math Wang et al. (2024b)), coding (HumanEval Chen et al. (2021a), HumanEval-Plus Liu et al. (2023), MBPP Austin et al. (2021), MBPP-Plus, MMLU-Coding, MMLU-Pro-coding), science (GPQA-Main Rein et al. (2023), GPQA-Diamond, SciBench Wang et al. (2024a), SciEval Sun et al. (2024), SciKnowEval Feng et al. (2024), MMLU-Sci, MMLU-Pro-Sci), medicine (MedMCQA Pal et al. (2022), MedQA Jin et al. (2021), PubMedQA Jin et al. (2019), MMLU-Med, MMLU-Pro-Med), and finance (FinanceBench Islam et al. (2023), FinQA Chen et al. (2021b), FPB Malo et al. (2014), MMLU-Finan, MMLU-Pro-Finan), where each dataset is randomly sampled up to 500 examples without replacement (except for SciKnowEval, from which we draw 800 instances to ensure sufficient coverage of its specialized tasks).

## G DATASET DETAILS AND SETTINGS

### G.1 DATASET DETAILS AND SETTINGS ON X-MAS-BENCH

**AIME-2024 Maxwell-Jia (2024)**. AIME-2024 comprises 30 challenging problems from the 2024 American Invitational Mathematics Examination (AIME), designed to evaluate advanced high school mathematical problem-solving skills. We sampled 30 examples in this dataset.

**AQUA-RAT Ling et al. (2017)**. AQUA-RAT is a large-scale dataset of approximately 100,000 algebraic word problems, each accompanied by natural language rationales, facilitating research in program induction and explainable AI. We sampled 254 examples in this dataset.

**GSM-Hard Gao et al. (2023)**. GSM-Hard is a more challenging variant of the GSM8K dataset, where numerical values are replaced with larger, less common numbers to test the robustness of mathematical reasoning in language models. We sampled 500 examples in this dataset.

**MATH Hendrycks et al. (2021b)**. The MATH dataset consists of 12,500 competition-level mathematics problems, each with detailed step-by-step solutions, aimed at evaluating and improving mathematical problem-solving abilities in AI systems. We sampled 250 examples in this dataset that were different from those in X-MAS-Design.

**MMLU Hendrycks et al. (2021a)**. The Massive Multitask Language Understanding (MMLU) benchmark includes multiple-choice questions across 57 subjects, assessing a model's world knowledge and problem-solving capabilities in zero-shot and few-shot settings. We sampled 500 examples in this dataset.

**MMLU-Pro Wang et al. (2024b)**. MMLU-Pro enhances the original MMLU by introducing more challenging, reasoning-focused questions and increasing answer choices from four to ten, thereby reducing the likelihood of correct guesses by chance. We sampled 500 examples in this dataset.

**HumanEval Chen et al. (2021a)**. HumanEval is a benchmark of 164 hand-written Python programming problems, each with a function signature, docstring, and unit tests, designed to evaluate the functional correctness of code generated by language models. We sampled 164 examples in this dataset.

**HumanEval-Plus Liu et al. (2023)**. HumanEval-Plus extends the original HumanEval by providing 80 times more test cases per problem, enabling a more rigorous assessment of code generation models' correctness and reliability. We sampled 164 examples in this dataset.

**MBPP Austin et al. (2021)**. The Mostly Basic Python Programming (MBPP) dataset comprises around 1,000 crowd-sourced Python programming tasks, each with a problem description, code solution, and test cases, targeting entry-level programming skills. We sampled 250 examples in this dataset that were different from those in X-MAS-Design.

**MBPP-Plus**. MBPP-Plus builds upon MBPP by significantly increasing the number of test cases per problem, offering a more stringent evaluation framework for assessing the correctness of code generated by language models. We sampled 361 examples in this dataset.

**GPQA-Main Rein et al. (2023)**. GPQA-Main is a dataset of 448 graduate-level, multiple-choice questions in biology, physics, and chemistry, crafted to be "Google-proof" and challenging for both humans and AI systems, thus serving as a benchmark for scalable oversight methods. We sampled 448 examples in this dataset.

**GPQA-Diamond**. GPQA-Diamond is an extension of the GPQA dataset, featuring even more challenging questions to further test the limits of AI models' scientific reasoning and knowledge without reliance on external resources. We sampled 198 examples in this dataset.

**SciBench Wang et al. (2024a)**. SciBench is a benchmark comprising college-level scientific problems sourced from instructional textbooks, designed to evaluate the complex reasoning, domain knowledge, and advanced calculation skills of large language models. We sampled 250 examples in this dataset that were different from those in X-MAS-Design.

**SciEval Sun et al. (2024)**. SciEval is a comprehensive benchmark with approximately 18,000 questions across chemistry, physics, and biology, assessing the capabilities of LLMs in basic knowledge, application, scientific calculation, and research ability. We sampled 500 examples in this dataset.

**SciKnowEval Feng et al. (2024)**. SciKnowEval evaluates large language models across five progressive levels of scientific knowledgestudying extensively, inquiring earnestly, thinking profoundly, discerning clearly, and practicing assiduouslyreflecting a holistic assessment inspired by ancient Chinese philosophy. We sampled 800 examples in this dataset.

**MedMCQA Pal et al. (2022)**. MedMCQA is a multiple choice question-answering dataset designed for medical domain evaluation, containing questions that assess a model's understanding of medical concepts and reasoning. We sampled 500 examples in this dataset.

