# OpenReview forum: "Diversity for The Win: Towards Building Multi-Agent Systems with Heterogeneous LLMs"
_ICLR.cc/2026/Conference — ICLR 2026 Conference Withdrawn Submission_

### Official Review · Reviewer_y77r · 2025-10-31

**Soundness:** 2
**Presentation:** 3
**Contribution:** 2
**Rating:** 4
**Confidence:** 3

**Summary:**

This paper explores the homogeneity limitation in multi-agent systems (MAS) based on large language models (LLMs), where all agents are driven by a single LLM, thereby constraining the collective intelligence of the system. To address this, a heterogeneous LLM-driven MAS paradigm is proposed, aiming to leverage the collective intelligence of diverse LLMs to enhance system performance. Specifically, X-MAS-Bench was constructed, evaluating 27 LLMs across 5 MAS-related functions and 5 common domains, with over 1.7 million evaluations conducted. Furthermore, based on experimental findings, it is demonstrated that transitioning from homogeneous to heterogeneous MAS design can significantly and consistently improve system performance.

**Strengths:**

1. This paper is well-written, with clear motivation and sound reasoning logic. Overall, it is highly comprehensible.
2. The X-MAS-Bench constructed in this paper systematically evaluates LLM capabilities from multiple domain-function perspectives, providing valuable insights for subsequent research and further validating the importance of the heterogeneous LLM-MAS paradigm.

**Weaknesses:**

1. X-MAS-Design relies on manual reference to X-MAS-Bench results for selecting the optimal LLM, which constitutes a simplistic and mechanistic approach.
2. The implementation of X-MAS-Design is predicated on X-MAS-Bench identifying the optimal LLM combination. For task domains beyond the benchmark's scope, additional costs will be required to determine the optimal combination, thereby undermining the method's value.

**Questions:**

1. There are multiple spelling errors in the text (e.g., "financespanning").
2.  The paper introduces the design of a heterogeneous LLM MAS. can it provide analysis and discussion regarding costs?
3. Could it provide a case study to demonstrate the LLM driver selection process in the heterogeneous LLM MAS design?

---

### Official Review · Reviewer_VkAd · 2025-11-01

**Soundness:** 3
**Presentation:** 2
**Contribution:** 3
**Rating:** 6
**Confidence:** 3

**Summary:**

This paper introduces X-MAS-Bench, which is used to evaluate different LLMs with different Multi-agent systems.  The paper evaluates 27 LLMs across multiply test set and find that using heterogeneous LLMs can significantly improve MAS's performance.

**Strengths:**

- The paper shows the performance of heterogeneous MAS systems through extensive experiments with different models and test sets. It also picks 5 representative MAS functions.
- To test X-MAS-Design, the authors chooses different MAS methods and tests on different domains, and the performance of X-MAS-Design is obviously and consistently higher than homogeneous MAS systems.

**Weaknesses:**

- X-MAS-Design picks “top performers” from X-MAS-Bench. While X-MAS-Bench covers a wide range of LLMs and domains and functions already, it cannot scale. For example, how would we decide if a new LLM is included?
- While it is understandable to only include open-source local models, it would be interesting to include some experiments with state of the art proprietary models. If a model is strong enough, do we still need to spend time picking heterogeneous LLMs?

**Questions:**

See weakness.

---

### Official Review · Reviewer_t6B1 · 2025-11-01

**Soundness:** 2
**Presentation:** 3
**Contribution:** 2
**Rating:** 2
**Confidence:** 3

**Summary:**

The authors introduce X-MAS, a framework and large-scale benchmark (X-MAS-Bench) for systematically evaluating MAS built from heterogeneous LLMs.
Results show that combining diverse LLMs often leads to higher accuracy, robustness, and complementary strengths than using a single model for all agents.

**Strengths:**

1. This paper is generally well-organized.

2. Provides a large-scale empirical study (X-MAS-Bench) evaluating 27 LLMs across 5 functional and 5 domain dimensions, with over 1.7 million evaluations.

3. Will releases code and data, promoting reproducibility and further research.

**Weaknesses:**

1. While the heterogeneous-LLM concept is valuable, the system mainly relies on model substitution rather than introducing a fundamentally new MAS design or coordination mechanism.

2. The overall contribution and scope of the paper are somewhat unclear. As an evaluation work, the number of evaluated models is quite limited; as a method paper, the proposed approach lacks sufficient novelty to stand out.

3. All three baselines are from October or May 2023, which makes them somewhat outdated given the rapid progress in multi-agent LLM research.

4. The reported performance improvements may largely stem from differences in individual model capabilities rather than from the effect of diversity itself. It remains unclear whether the same gains would persist if all constituent models were comparably strong. A controlled experiment with equally capable LLMs would help isolate the true impact of heterogeneity.

5. The ablation results are somewhat confusing. The finding that model combinations must be carefully selected based on extensive prior evaluations seems to suggest that the observed performance gains stem mainly from differences in individual model capabilities rather than from the proposed framework design itself. Moreover, this approach appears non-generalizable and impractical—if each new task requires large-scale evaluations to identify an effective heterogeneous design, the method would be too costly to deploy in practice.

**Questions:**

Please refer to the weaknesses section for main questions. More minor questions are given below:

The description of X-MAS-Proto is quite confusing. From the paper, it seems that X-MAS-Proto is treated as one of the MAS frameworks (similar to LLM-Debate or DyLAN), but its exact nature is unclear. Could the authors clarify whether X-MAS-Proto itself is a MAS framework? It looks like it can be applied in both homogeneous and heterogeneous settings, though I’m not entirely sure if I’m interpreting this correctly.

Additionally, X-MAS-Proto is first mentioned around line 312 with the phrase “The system (see the MAS in Figure 1)”, but it does not appear explicitly in Figure 1, making it difficult to connect the description to the depicted architecture. I suspect that X-MAS-Proto corresponds to the upper-right structure in Figure 1, though this is never clearly stated in the text. A more explicit and self-contained introduction to X-MAS-Proto earlier in the paper would greatly improve clarity.

---

### Official Review · Reviewer_SHSH · 2025-11-09

**Soundness:** 1
**Presentation:** 2
**Contribution:** 1
**Rating:** 2
**Confidence:** 4

**Summary:**

This work studies how task performance is influenced when adopting heterogeneous LLMs within multi-agent systems. The authors perform a large-scale empirical analysis and find that heterogeneous configurations consistently outperform homogeneous ones. Their results indicate that no single model is universally optimal; instead, combining diverse LLMs can exploit complementary strengths across domains and functions, leading to performance gains in both chatbot-only and mixed chatbot-reasoner settings.

**Strengths:**

- The problem of studying optimizing multi-agent system is important in the modern context.

**Weaknesses:**

- The writing could be improved to more closely follow standard academic structure. Currently, citations are embedded directly in the text, which makes the narrative hardly readable (e.g., Experimental setups. line 240-255)
- For a benchmark paper, the presented MAS structures seem limited, and the extent of data diversity is not clearly articulated.
- Several closely related works with similar problem setups and benchmarking goals are not cited or discussed, such as LLMSelector (“Optimizing Model Selection for Compound AI Systems”), Optimas (“Optimas: Optimizing Compound AI Systems with Globally Aligned Local Rewards”), and Dylan ("A Dynamic LLM-Powered Agent Network for Task-Oriented Agent Collaboration") among others in recent literature.
- The novelty of the benchmarks does not seem to be convincing, given the setting is a bit simplified or in similar complicity compared to the existing works.

Minor comments:
- The term “system intelligence” could be defined more clearly. For example, in the introduction you state: “This manner inherently limits the systems intelligence to that of the underlying model.” It would be helpful to clarify what aspects of intelligence this refers to.
- There are a few minor typos (e.g., `˘a` on line 199).

**Questions:**

- How exactly does this work differ from the datasets used in LLMSelector and Optimas and Dylan?

---

### Note · Authors · 2026-01-09

I have read and agree with the venue's withdrawal policy on behalf of myself and my co-authors.